# Diet Quality and Neighborhood Environment in the Atlantic Partnership for Tomorrow’s Health Project

**DOI:** 10.3390/nu12103217

**Published:** 2020-10-21

**Authors:** Kaitlyn Gilham, Qianqian Gu, Trevor J. B. Dummer, John J. Spinelli, Rachel A. Murphy

**Affiliations:** 1School of Population and Public Health, University of British Columbia, Vancouver, BC V6T 1Z3, Canada; kaitlyn.gilham@alumni.ubc.ca (K.G.); paulinggqq23@gmail.com (Q.G.); trevor.dummer@ubc.ca (T.J.B.D.); jspinelli@bccrc.ca (J.J.S.); 2Cancer Control Research, BC Cancer, Vancouver, BC V5Z 1L3, Canada

**Keywords:** healthy eating, dietary patterns, food environment, Atlantic Partnership for Tomorrow’s Health (Atlantic PATH)

## Abstract

An understanding of relationships between different constructs of the neighbourhood environment and diet quality is needed to inform public health interventions. This study investigated associations between material deprivation, social deprivation and population density with diet quality in a cohort of 19,973 Canadian adults aged 35 to 69 years within the Atlantic PATH cohort study. Diet quality, a metric of how well diet conforms to recommendations was determined from a 24-item food frequency questionnaire. Neighbourhood environment data were derived from dissemination area level Census data. Two deprivation indices were evaluated: material and social deprivation, which reflect access to goods and amenities and social relationships. Multi-level models were used to estimate relationships (mean differences and 95% CI) between neighbourhood environment and diet quality, adjusting for covariates. Mean diet quality was lower in the most socially deprived neighbourhoods compared to the least socially deprived: −0.56, 95% CI (−0.88, −0.25). Relationships between diet quality and population density differed between urban and rural areas (*p*-interaction < 0.0001). In rural areas, diet quality was higher in intermediate-density neighbourhoods: 0.54, 95% CI (0.05, 1.03). In urban areas, diet quality was lower in intermediate-density and the most-dense neighbourhoods: −0.84, 95% CI (−1.28, −0.40) and −0.72, 95% CI (−1.20, −0.25). Our findings suggest socially deprived and high-density neighbourhoods are associated with lower diet quality and possible urban-rural differences in neighbourhood environment-diet quality relationships. Additional studies are needed to determine the temporal nature of relationships and whether differences in diet quality are meaningful.

## 1. Introduction

A healthy diet is a critical component of overall health, including achieving and maintaining a healthy body weight and prevention of chronic diseases such as obesity, diabetes, cardiovascular disease and cancer [1]. Although, definitions vary, generally a healthy dietary pattern includes a variety of vegetables, fruits, grains, at least half of which are whole grains, fat-free or low-fat dairy, a variety of protein foods, oils and limited saturated fats, trans fats, added sugars and sodium [2]. Over the past two decades, unhealthy dietary patterns-those with excess consumption of nutrient-poor and energy-dense foods has increased, resulting in a global decline in diet quality [3].

Dietary intake and food selection are highly personal behaviours influenced by sociodemographic, lifestyle and health-related factors [4]. The neighbourhood environment where people live, work and engage in recreation may also influence eating behaviour, as may the food environment, which encompasses the accessibility and affordability of foods in the built environment [5]. Lower diet quality has been shown to be associated with greater deprivation within a community [6,7,8]. For example, poorer adherence to national Dietary Guidelines was observed in the most deprived areas relative to the least deprived areas in an Australian study [7]. Studies in the United States reported that high neighborhood socioeconomic status (SES) was associated with greater access to supermarkets, less access to fast food restaurants and a healthier diet [9,10]. Conversely, several studies in Canada reported increased availability of both healthy and unhealthy food retailers in more deprived areas and no association between neighbourhood deprivation and diet quality [11,12,13].

Most studies have investigated relationships between diet and the neighbourhood environment using broad definitions of the socioeconomic environment. However, the socioeconomic environment is multi-faceted, reflecting single parenthood, education, marital status, unemployment rate, rented private dwellings and the average value of dwellings. Given this, studies have conceptualized deprivation as two independent dimensions: material and social, which may have unique relationships with dietary intake. Material deprivation represents deprivation of goods and conveniences, such as housing and cars, while social deprivation reflects deprivation of social networks, from family to community [14]. The nuances of deprivation may not have been captured in prior studies as generally they rely on material deprivation or do not consider the two dimensions individually [11,15,16,17,18]. Additional information on the relationship between diet and neighbourhood environment is critical to informing strategies to support healthy eating and potentially, clarify inconsistent relationships previously identified.

The aim of this study was to assess relationships between multiple indices of the neighbourhood environment: material deprivation, social deprivation, population density with diet quality in a large cohort of Canadian adults.

## 2. Materials and Methods

### 2.1. Study Sample

Participants were drawn from the Atlantic Partnership for Tomorrow’s Health (Atlantic PATH) study, a cohort of 31,173 people aged 35 to 69 in Nova Scotia, New Brunswick, Prince Edward Island and Newfoundland and Labrador between 2009 and 2015 [19]. Participants were enrolled between 2009 and 2015 as part of a larger pan-Canadian effort; the Canadian Partnership for Tomorrow’s Health Project (CanPath, formerly the Canadian Partnership for Tomorrow Project) [20]. Details of recruitment have been published previously [19].

Data for this study, was drawn from a prior study of dietary quality among cancer survivors and those without cancer [21]. The analytical sample of 19,973 participants includes those with complete information on diet quality, neighbourhood environment, census data for dissemination areas (DAs) and income information from the voluntary NHS, as described previously [21]. The study was reviewed and approved by the Behavioural Research Ethics Board at the University of British Columbia (Certificate #H15-02854).

### 2.2. Dietary Intake

Participants reported their usual dietary intake using a short, semi-quantitative food frequency questionnaire (FFQ) developed for the Atlantic PATH cohort. The FFQ was designed to capture consumption reflecting the national dietary guidelines at the time of the study (Eating Well with Canada’s Food Guide). The set of 24 questions assessed the daily intake of 60 food items or groups including fruit and vegetables, grains, dairy products, meat and alternatives, snacks, desserts, non-diet soft drinks, fats, sauces and salt in the prior 12 months. Participants were asked to indicate the number of servings they consumed in a typical day for four main food groups: fruits and vegetables, grain products, dairy products and meat and alternatives. For other foods, participants were asked to indicate the usual frequency of consumption using times per day, times per week, times per month or rarely/never.

The Atlantic PATH diet quality score was developed following methods of the American and Canadian Health Eating Index (HEI)s whose scoring schemes reflect national dietary guidelines and population-based studies for disease prevention [22,23]. Intake of food from four food groups (fruits and vegetables, grains, dairy products, meat and alternatives) are assigned a score ranging from 0 to 10. A score of 10 indicates meeting dietary guidelines for a given food group (e.g., ≥2 servings/day of milk and dairy products for males and females aged ≤50 or ≥3 servings/day for males and females aged >50). Intake of foods to consume in moderation (i.e., snacks, desserts and non-diet drinks) and dietary behaviors are assigned a score of 1 for meeting or 0 for not meeting a given recommendation. For example, the component ‘make at least half of grain products whole grain each day’ receives a score of 1 if a participant’s ratio of whole grain to total grain product intake/day is ≥0.5. The components are then summed to generate an overall diet quality score from 0 to 60. Higher scores indicate better diet quality. The diet quality score has been shown to have content and construct validity and acceptable internal reliability (Cronbach’s alpha 0.74) [21]. Additional details of the score derivation in the Atlantic PATH have been previously published [24].

### 2.3. Neighbourhood Environment

Population density and area-level deprivation were calculated from the 2011 census and the National Household Survey (NHS), using data at the DA level [25]. DAs are the smallest standard geographic area for which Census data is disseminated in Canada. DAs contain between 400 to 700 persons and are composed of one or more adjacent dissemination block (an area equivalent to a city block). Neighbourhood environment characteristics were linked to each individual by postal codes using the Postal Code Conversion File Plus Version 6C (Statistics Canada, Ottawa, Ontario, Canada) [26]. Individuals living in the same DAs were assigned the same values of neighbourhood environment characteristics. Population density for each DA was calculated by dividing the population count by the land area, expressed in residents per km^2^. Area was classified as urban or rural (urbanicity), whereby urban areas referred to any census metropolitan areas or census agglomerations with a core population of at least 10,000 [26].

Socioeconomic deprivation was constructed using six indicators from the Census: proportion of people without a high school diploma, average individual income, employment rate, proportion of single-parent families, proportion of people living alone and proportion of people who were separated, divorced or widowed [14,27,28]. Indicators were standardized to the age and sex structure of the population aged 15 years or older in the Atlantic provinces in 2011 and normalized using Tukey’s transformation. Principal component analysis was used to reduce the indicators into a smaller set of variables and identify deprivation indices [29]. An orthogonal varimax rotation was used. Components with an eigenvalue above one were retained. Marital status, the proportion of single-parent families and the proportion of people living alone loaded on the first component (referred to as “social deprivation”), while education, income and employment loaded on the second (referred to as “material deprivation”). Factor scores for the two components were calculated as material and social deprivation indices for all DAs in the Atlantic provinces (*n* = 4142). DAs were then categorized into tertiles based on material deprivation and social deprivation.

### 2.4. Covariates

Potential covariates were obtained from the Atlantic PATH baseline questionnaire which included age, sex, ethnicity, education, marital status, household income, smoking status, alcohol consumption, physical activity, diagnosis of prevalent diabetes and myocardial infarction, urbanicity and province of residence. Age was categorized into seven groups (35–39, 40–44, 45–49, 50–54, 55–59, 60–64 and 65–69 years old). Ethnicity was dichotomized as white and non-white. Education level was categorized as high school or lower, college level, university or higher. Marital status was dichotomized as living with or without partners. Household income (Canadian dollars) was categorized as less than $25,000, $25,000–49,999, $50,000–74,999, $75,000–150,000 and more than $150,000. Physical activity was evaluated using the International Physical Activity Questionnaire (IPAQ) [30] and categorized as low, moderate and high [31]. Former smokers were those who had smoked at least 100 cigarettes during their lifetime but did not smoke within the past 30 days of the survey. Current smokers were those who had smoked at least 100 cigarettes in their lifetime and smoked in the past 30 days, either daily (regular smokers) or not daily (occasional smokers). All other participants were categorized as non-smokers. Participants were classified as abstainers (never drinking alcohol), former (no alcohol over the past 12 months), occasional (≤2–3 times/month), regular (≥once/week but ≤2–3 times/week) and habitual drinkers (≥4–5 times/week). Self-reported diagnosis of diabetes and myocardial infarction were recorded as “yes” or “no.”

### 2.5. Statistical Analysis

Multiple imputation was used to handle missing covariate data with the Multivariate Imputations by Chained Equation (MICE) package in R [32]. Education, household income, marital status, ethnicity, smoking status, alcohol consumption, physical activity, myocardial infarction and diabetes were imputed. Age, sex, body mass index (BMI) calculated from self-reported height and weight, self-perceived health and diet quality were included as auxiliary variables in the multiple imputation [33]. A fully conditional specification (FCS) method was adopted to impute different types of data using separate conditional distributions. Forty imputed datasets were created with 200 iterations were performed before reaching convergence [32]. The distributions of continuous variables were examined using means, standard deviations, medians and ranges. Counts and frequencies were calculated for categorical variables.

Mixed-effect models were conducted due to the nested data structure, in which individuals are nested within forward sortation areas (FSAs). Because the number of participants was too few within DAs, FSAs were selected, which are defined by the first three digits of the postal code, as the level of clusters in the multi-level models to achieve sufficient sample sizes. In this study, the number of participants within FSAs ranged from 1 to 534, with an average of 79. Linear mixed-effect models (LMMs) were used to provide estimates of the association with diet quality as a continuous variable. A random intercept was included in the models to consider the area-level variation in diet quality. Assumptions of linearity in models were confirmed through residual plots that showed random dispersion [34].

Potential confounders were assessed using backward elimination, including age, sex, household income, education, marital status, ethnicity, smoking status, alcohol consumption, physical activity, BMI, diabetes, myocardial infarction, social deprivation, material deprivation, population density, urbanicity and province of residence. At each step, the least significant variable was removed first until all variables in the model had a *p*-value < 0.20 [35]. The neighborhood characteristics, material deprivation, social deprivation and population density were analyzed in tertiles as the primary independent variables in separate models predicting diet quality. The least deprived and least dense tertiles were the referent. Significant confounders identified with backward elimination included age, sex, household income, education, ethnicity, smoking status, alcohol consumption, physical activity, cancer, diabetes and province of residence, which were included in all models. 

Individual SES (income and education) and urbanicity were examined as potential effect modifiers of diet quality. All data analyses were performed in R version 3.4.2 (R Foundation for Statistical Computing, Vienna, Austria) and significance level of 0.05 was used.

## 3. Results

Participants’ demographic, socioeconomic, lifestyle, health-related and residential characteristics are presented in Table 1. The sample had a higher percentage of females (69.1%) compared to males, whites (93.2%) compared to non-whites and people living with partners (80.9%) compared to those living without partners. Most participants had a high SES—over 50% earned an annual household income greater than $75,000, while only around 5% had less than $24,999. There were 40.7% and 40.1% of participants with a college degree and Bachelor’s degree or above, respectively. Most participants had a healthy lifestyle with regard to smoking and physical activity. Fewer than 10% of participants currently smoked cigarettes at the time of the survey, while 50.8% had never smoked and 39.3% were former smokers. Approximately 50% and 30% reported high and moderate levels of physical activity, respectively. Most participants were current drinkers at the time of the survey. The prevalence of diabetes and myocardial infarction was 5.1% and 1.8% in the analytic sample. The majority of participants (71.8%) were from urban areas.

The mean diet quality of all participants was 38.8 out of 60 (SD 8.65), suggesting modest adherence with dietary recommendations. With respect to the individual diet quality components, the highest score, representing greater adherence to serving recommendations was meat and alternatives (mean: 8.95 out of 10), while the lowest score was grain products (mean: 4.32 out of 10). The majority of participants consumed more whole fruit and vegetables than juice (94.0%), drank low-fat milk or milk alternatives (85.2%) and 81.5% reported at least half of grain products consumed were whole grains. Mean diet quality ranged from a low of 38.3 (SD 8.86) and 38.3 (SD 8.97) in the most materially deprived environment and most socially deprived environment to a high of 39.2 (SD 8.39) in the least socially deprived neighbourhood (Table 2).

Nearly half of participants (*n* = 9076, 45.4%) were from the least materially deprived areas in the Atlantic provinces, while 4532 (22.7%) were from the most materially deprived areas.

With respect to social deprivation, 36.5% were from the least socially deprived, 30.5% from the most deprived and 33.0% from areas with an intermediate level of social deprivation. In addition, 7822 (39.2%) participants were from densely populated areas and 5519 (27.6%) were from the least dense areas.

### 3.1. Relationships between Material Deprivation and Diet Quality

Compared to residents of the least materially deprived areas, those living in areas with intermediate and the highest level of material deprivation had lower mean diet quality: mean difference −0.60 (95% CI: −0.90, −0.30) and −0.85 (95% CI: −1.20, −0.49), respectively (Table 3).

However, no significant association was found between material deprivation and diet quality, after adjustment for individual sociodemographic factors, lifestyle behaviors, health-related characteristics, social deprivation, population density, urbanicity and province of residence.

### 3.2. Relationships between Social Deprivation and Diet Quality

Compared to the least socially deprived areas, there was a decreasing trend in the mean diet quality of people living in areas with intermediate and the highest level of social deprivation (intermediate deprivation: −0.41, 95% CI: −0.71, −0.11; most deprived: −0.87, 95% CI: −1.20, −0.54) (Table 4).

After adjusting for individual-level confounders as well as urbanicity and province of residence, the highest level of social deprivation remained significantly and inversely associated with high diet quality: −0.56 (95% CI: −0.88, −0.25). Individual SES (income and education) and urbanicity did not modify associations.

### 3.3. Relationships between Population Density and Diet Quality

Differences in mean diet quality scores did not reach significance across areas of different population density in unadjusted models (Table 5). After adjusting for individual-level sociodemographic, lifestyle and health-related factors as well as social deprivation and province of residence, people living in the most densely populated areas had lower mean diet quality by 0.39 units (95% CI: −0.77, −0.01) than people living in the least dense areas.

Differences in diet quality by population density were more evident in urban areas than in rural areas (Table 6). In urban areas, living in the densest areas was associated with a lower mean diet quality by 0.72 units (95% CI: −1.20, −0.25). In contrast, in rural areas, higher population density was associated with better diet quality. Compared to those in the least dense population, mean diet quality was higher by 0.54 (95% CI: 0.05, 1.03) in areas of intermediate population density but no difference was observed for those in the most dense areas.

## 4. Discussion

Findings from this large study of Atlantic Canadians suggest that people who live in neighborhood environments characterized by greater material deprivation, social deprivation and population density may have lower diet quality. Some relationships, such as those with material deprivation were attenuated with adjustment for individual-level SES, sociodemographic and health related factors. The findings also suggest possible differences in the association between diet quality and population density between urban and rural neighborhoods. In urban areas, diet quality was lower in the most densely populated neighborhoods, whereas in rural areas, diet quality was lower in the least densely populated neighborhoods. However, while statistically significant, it is unclear whether these small differences in diet quality are meaningful in terms of health. A difference in diet quality of 0.72 (i.e., the least dense and most dense urban populations) can be equated to an additional 4.5 servings of fruits/vegetables per week. While this aligns with public health guidance to ‘start small’ [36], the impact on overall health is unclear. Parallels, however, can be drawn from dose response studies which suggest benefits of for increments of even one serving of fruit and/or vegetables per day [37,38]. For instance, a meta-analysis of 16 prospective cohorts reported a pooled hazard ratio for all-cause mortality of 0.95 (95% CI 0.92, 0.98) and cardiovascular mortality (hazard ratio 0.96, 95% CI 0.92, 0.99) for increments of one serving/day of fruit and vegetables [38].

Evidence on neighborhood deprivation and dietary intake is inconsistent; some studies have indicated lower diet quality in more deprived areas while others suggested no association [7,8,13]. Two large-scale Australian studies, one of which was based on a nationally representative sample, showed that people living in more deprived areas had poorer adherence to the Australian dietary guidelines [7,8]. A previous study in Alberta, Canada, however, showed no significant association between neighborhood deprivation and a HEI derived from Canadian dietary guidelines, with and without adjustment for individual-level SES and food environment [13]. A systematic review of associations between health behaviors and neighborhood deprivation found no clear results in regard to fruit and vegetable consumption, likely due to heterogeneous definitions of neighborhood deprivation [39].

Differences between our findings and prior studies could reflect heterogeneity in study methods in terms of statistical adjustment, definitions of neighborhood deprivation and dietary assessment or between study geographical differences. For example, Backholer et al. [7] reported that individual-level education had a larger effect on diet quality than area-level deprivation [7,8]. Lack of consideration for possible confounders in prior studies also may have obscured true relationships. Differences in the definition of neighborhood deprivation make it difficult to compare results. The majority of studies use a composite score, which is mainly based on education, income and employment [7,8,13,39], versus the definition used herein which considers relationships with material and social deprivation separately.

The modifying effect of urbanicity on population density and diet quality but not social or material deprivation is intriguing. As outlined above, there are few studies to draw comparisons to and ecological perspectives on dietary intake in Canada, which has unique individual, social, environmental and public policies are even fewer [40]. Parallels can be drawn to the SPOTLIGHT study which was conducted in five urban European regions [41]. They reported that residents in neighborhoods with a low residential density and median income had lower vegetable consumption compared to residents in neighborhoods with high residential density and median income. We did not have access to information on food outlets such as fast-food outlets, convenience stores or grocery stores within neighborhoods, which may have provided important insight into urban/rural differences. A study in Nova Scotia (where ~64% of our population resided) reported higher prevalence of fast food outlets in urban areas [42]. It is therefore possible that densely populated urban areas in our study had greater access to fast food or non-nutritious food (food swamps) while less dense rural areas may represent food deserts; areas with poor access to nutritious food [40]. Incorporation of data on food outlets into future studies of structural environments and dietary intake would help to provide context to findings. It is also important to describe the geographical region of the Atlantic Provinces to provide context to our findings and details which will help future studies determine if our findings are applicable to them. In 2012, the total population of the Atlantic Provinces was ~2.3 million [43]. We applied Canadian Census criteria to define urban areas (populations exceeding 10,000), of which there are 20 in the Atlantic Provinces, with the largest metropolitan area (Halifax) comprising ~400,000 people.

A strength of this study is the large sample of participants from diverse neighborhoods in all Atlantic provinces, which increases the generalizability of findings to this region. The statistical models, which considered the potential geographical clustering of observations is also a strength. A further strength was consideration of diet quality; how well an individual follows Canadian dietary recommendations, versus focusing on single foods (e.g., fruits and vegetables) which may be more reflective of overall health. However, comparability with other studies is limited by the diet assessment tool which was developed specifically for the Atlantic PATH cohort. Future research which draws on more widely used diet quality measures such as the HEI [23] would facilitate comparisons, although such an undertaking which requires a more in depth dietary tool(s) in a large population like the Atlantic PATH would be substantial. There are also inherent errors (e.g., underreporting) in self-report intake assessments that may have biased our results toward the null. Although, measurement of diet components have less error than estimates of absolute energy intake [44], which we did not consider in our study. The cross-sectional study design makes it difficult to infer temporal relationships and causality and the use of DAs to estimate neighborhoods. This approach may not capture where people engage in daily activities (e.g., work, school, recreation) which may also influence dietary behavior. The methodology used to measure material and social deprivation has strengths and limitations. The Pampalon index, was based on Townsend’s widely accepted definition of deprivation [14] and has been widely used to monitor social inequalities in Canada [45,46]. Conversely, the 2011 NHS was used as one of the sources to determine neighborhood deprivation [47]. The NHS is voluntary and non-response may be more likely among people with lower or higher household income. However, the NHS was the most representative source available during the study period.

## 5. Conclusions

Diet quality was low overall in this large population of adults in Atlantic Canada, providing evidence of the broad need for public health interventions to improve healthy eating. Neighborhoods characterized by higher levels of social deprivation and material deprivation may be particularly in need of effective and sustainable solutions to improve dietary intake. The findings also highlight the importance of considering potential urban/rural differences when studying relationships between the built environment and dietary intake. However, more research is needed to address gaps and inconsistencies in the literature. A deeper understanding of mechanisms underlying associations between neighborhood characteristics and diet quality is also needed to inform public health approaches to mitigate diet and health inequity.

## Figures and Tables

**Table 1 nutrients-12-03217-t001:** Characteristics of the analytic sample from the Atlantic PATH cohort.

Characteristics	Analytic Sample
	*N* = 19,973
	*N* (%)
Age	
35–39 years	2007 (10.1)
40–44 years	2369 (11.9)
45–49 years	3056 (15.3)
50–54 years	3549 (17.8)
55–59 years	3608 (18.1)
60–64 years	3322 (16.6)
65–69 years	2062 (10.3)
Sex	
Female	13,797 (69.1)
Male	6176 (30.9)
Marital status	
Living without partners	3816 (19.2)
Living with partners	16,115 (80.9)
Ethnicity	
White	17,418 (93.2)
Non-white	1279 (6.84)
Income	
$0–24,999	852 (4.52)
$25,000–49,999	3213 (17.1)
$50,000–74,999	3973 (21.1)
$75,000–149,999	8302 (44.1)
$>150,000	2495 (13.3)
Education (completed)	
≤High school	3836 (19.3)
College	8095 (40.7)
≥Bachelor’s degree	7975 (40.1)
Smoking status ^‡^	
Non-smoker	10,067 (50.8)
Former	7775 (39.3)
Occasional	500 (2.52)
Regular	1460 (7.37)
Alcohol consumption ^§^	
Abstainer	830 (4.18)
Former	1332 (6.71)
Occasional	8279 (41.7)
Regular	6254 (31.5)
Habitual	3170 (16.0)
Physical activity	
Low	3979 (20.8)
Moderate	5876 (30.7)
High	9296 (48.5)
BMI	
Normal	3994 (30.9)
Underweight	77 (0.60)
Overweight	4922 (38.0)
Obese	3947 (30.5)
Diabetes	
Yes	998 (5.05)
No	18,781 (5.0)
Myocardial infarction	
Yes	353 (1.78)
No	19,489 (98.2)
Urbanicity	
Urban	14,347 (71.8)
Rural	5626 (28.2)
Province	
Newfoundland/Labrador	2512 (12.6)
Prince Edward Island	377 (1.89)
Nova Scotia	12,571 (62.9)
New Brunswick	4513 (22.6)

^‡^ Non-smoker: has never smoked, former: has smoked at least 100 cigarettes before but not within the past 30 days, occasional: smoked at least once within the past 30 days but not daily, regular: smoked daily. All other participants were categorized as non-smokers. ^§^ Abstainer: has never drunk alcohol, former: has drunk alcohol before but not over the past 12 months, occasional: drank ≤2–3 times/month over the past 12 months, regular: drank ≥ once/week but ≤2–3 times/week, habitual drinkers: drank ≥4–5 times/week. PATH: Partnership for Tomorrow’s Health, BMI: body mass index.

**Table 2 nutrients-12-03217-t002:** Percentage of participants with high diet quality and mean diet quality scores by categories of neighbourhood environment.

	Overall Analytic Sample	Diet Quality Score	
	*N* = 19,973	*N* = 19,973	
	*N* (%)	Mean	SD
Material Deprivation			
Least deprived	9076 (45.4)	39.1	8.55
Intermediate	6365 (31.9)	38.7	8.62
Most deprived	4532 (22.7)	38.3	8.86
Social Deprivation			
Least deprived	7287 (36.5)	39.2	8.39
Intermediate	6590 (33.0)	38.8	8.61
Most deprived	6096 (30.5)	38.3	8.97
Population Density			
Least dense	5519 (27.6)	39.0	8.42
Intermediate	6632 (33.2)	38.9	8.68
Most dense	7822 (39.2)	38.5	8.78

SD: standard deviation.

**Table 3 nutrients-12-03217-t003:** Mean difference in diet quality (95% CI) of participants living in more materially-deprived areas versus least deprived areas, results from unadjusted and adjusted linear mixed-effect models (LMMs).

	Unadjusted	Adjusted ^†^
Material Deprivation		
Least deprived	reference	reference
Intermediate	−0.60 (−0.90, −0.30) *	−0.25 (−0.54, 0.05)
Most deprived	−0.85 (−1.20, −0.49) *	−0.15 (−0.51, 0.21)

* *p* < 0.05, **^†^** Adjusted for age, sex, household income, education, ethnicity, alcohol consumption, physical activity, cancer, diabetes, population density, social deprivation, urbanicity, province of residence.

**Table 4 nutrients-12-03217-t004:** Mean difference in diet quality score (95% CI) of participants living in more socially-deprived areas versus least deprived areas, in unadjusted and adjusted LMMs.

	Unadjusted	Adjusted ^†^
Social Deprivation		
Least deprived	reference	reference
Intermediate	−0.41 (−0.71, −0.11) *	−0.28 (−0.57, 0.01)
Most deprived	−0.87 (−1.20, −0.54) *	−0.56 (−0.88, −0.25) *

* *p* < 0.05, ^†^ Adjusted for age, sex, household income, education, marital status, ethnicity, alcohol consumption, physical activity, cancer, diabetes, urbanicity and province of residence.

**Table 5 nutrients-12-03217-t005:** Mean difference in the diet quality score (95% CI) of participants living in more densely populated areas versus the least populated areas, results from unadjusted and adjusted LMMs.

	Unadjusted	Adjusted ^†^
Population Density		
Least dense	reference	reference
Intermediate	−0.24 (−0.58, 0.09)	−0.28 (−0.60, 0.04)
Most dense	−0.31 (−0.69, 0.07)	−0.39 (−0.77, −0.01) *

* *p* < 0.05, ^†^ Adjusted for age, sex, household income, education, marital status, ethnicity, smoking status, alcohol consumption, physical activity, cancer, diabetes, social deprivation, province of residence.

**Table 6 nutrients-12-03217-t006:** Mean difference in diet quality (95% CIs) among participants living in more densely populated areas versus the least populated areas, results from adjusted LMMs stratified by urbanicity.

Urbanicity	Population Density	Mean Difference in Diet Quality (95% CI) ^†^	*p* _interaction_ ^§^
Urban	Least dense	reference	<0.001
	Intermediate	−0.84 (−1.28, −0.40) *	
	Most dense	−0.72 (−1.20, −0.25) *	
Rural	Least dense	reference	
	Intermediate	0.54 (0.05, 1.03) *	
	Most dense	−0.18 (−0.95, 0.58)	

* *p* < 0.05, ^†^ LMM adjusted for age, sex, household income, highest education, marital status, ethnicity, smoking, alcohol consumption, physical activity, diabetes, cancer, diabetes, social deprivation, urbanicity, province of residence; ^§^
*p*-value for the interaction between population density and urbanicity in the adjusted LMM.

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
