# Peer review of "Diet Quality and Neighborhood Environment in the Atlantic Partnership for Tomorrow’s Health Project"

_nutrients, 2020, doi:10.3390/nu12103217_

Round 1
Reviewer 1 Report
The current manuscript describes an interesting evaluation of an existing dataset on overall dietary habit in relation to two estimates of deprivation. This approach is novel
- Line 171 and elsewhere - cross-references to Tables in the descriptive text of your Results do not appear to be displaying correctly. Please check and/or update original document.
- Lines 261-262 - in relation to the point raised, it may be worth considering whether previous estimates on frequency of fruit and vegetable consumption per serving in terms of e.g. cardiovascular risk could be considered to think about the potential to impact long-term health outcomes. This would likely be most relevant if you can find dose-response estimates from previous observational studies in similar populations though.
- Lines 297- 305 - a major consideration that should be considered in relation to this manuscript is the inherent strengths and/or weaknesses of the methods used to estimate both types of deprivation. Please consider expanding this section.
- Discussion/Conclusions - additional information suggesting the types of future research that are necessary to expand knowledge in this field should be included.
Author Response
Reviewer 1
The current manuscript describes an interesting evaluation of an existing dataset on overall dietary habit in relation to two estimates of deprivation. This approach is novel
- Line 171 and elsewhere - cross-references to Tables in the descriptive text of your Results do not appear to be displaying correctly. Please check and/or update original document.
-This was an error in the pdf conversion as it displayed correctly in the word document. We have corresponded with the editorial staff and believe this has now been resolved.
- Lines 261-262 - in relation to the point raised, it may be worth considering whether previous estimates on frequency of fruit and vegetable consumption per serving in terms of e.g. cardiovascular risk could be considered to think about the potential to impact long-term health outcomes. This would likely be most relevant if you can find dose-response estimates from previous observational studies in similar populations though.
-Thank you for the helpful suggestion. We have added the following to the text “Parallels, however, can be drawn from dose response studies which suggest benefits of for increments of even one serving of fruit and/or vegetables per day(35,36). For instance, a meta-analysis of 16 prospective cohorts reported a pooled hazard ratio for all-cause mortality of 0.95 (95% CI 0.92, 0.98) and cardiovascular mortality (hazard ratio 0.96, 95% CI 0.92, 0.99) for increments of one serving/day of fruit and vegetables (36).“
- Lines 297- 305 - a major consideration that should be considered in relation to this manuscript is the inherent strengths and/or weaknesses of the methods used to estimate both types of deprivation. Please consider expanding this section.
-We have expanded this as suggested “The methodology used to measure material and social deprivation has strengths and limitations. The Pampalon index, was based on Townsend’s widely accepted definition of deprivation (41)and has been widely used to monitor social inequalities in Canada (42,43). Conversely, the 2011 NHS was used as one of the sources to determine neighbourhood deprivation (44). The NHS is voluntary, and non-response may be more likely among people with lower or higher household income. However, the NHS was the most representative source available during the study period.”
- Discussion/Conclusions - additional information suggesting the types of future research that are necessary to expand knowledge in this field should be included.
-We have included future directions in a number of places in this discussion and conclusion. For example “Future research which draws on a more widely used tool such as the HEI (41)would facilitate comparisons, although such an undertaking which requires a more in depth diet assessment in a large population like the Atlantic PATH would be substantial.” And “A deeper understanding of mechanisms underlying associations between neighbourhood characteristics and diet quality is also needed to inform public health approaches to mitigate diet and health inequity.“

Reviewer 2 Report
GENERAL COMMENTS
This is a well-written paper on an interesting subject. My main concern is that it looks as though the differences observed in diet quality score are really small. However we don’t know what the scores actually mean. We do know that the maximum is 60. However virtually all of the averages hover around 38-9 with a standard deviation of around 8.5 (Table 2.). Authors need to do a better job of putting their diet quality measure into meaningful context. It is not easy to draw comparisons with other studies based on this unique measure. So, the onus is upon them to explain this clearly. They also need to put this into context in other ways. Canada is a unique country and the maritime provinces (and Newfoundland Labrador) are really unusual in a lot of ways. For example, there are no really large cities in the region. I believe that Halifax is the largest city with a population of about 400,000. Still, with a total population of around 2 million in a really large area it may be classified as fairly urban! So, generalizing to places with huge cities and lots of urban blight may be difficult. It seems that this may be the underlying cause of what I perceived to be unintuitive results.
SPECIFIC COMMENTS
Abstract
Line 15 – A 24-item FFQ seems very short
Lines 20-1 – I am not sure what diet quality means in this context. Of course, this applies to text on the lines 23-5 as well.
Lines 25-6 – Insert hyphen in compound adjective “high-density”
Introduction
Line 34 – Some might argue that diet is important for many things over and above body weight and body weight itself is determined by physical activity energy expenditure to a large extent.
Line 39 – Insert hyphen in compound adjectives “nutrient-poor” and “energy-dense”
Line 41 – I don’t see the distinction between dietary intake and food choices.
Lines 42-3 – It seems that there are more than one neighborhood environments that must be considered. People may not live where they work or engage in physical activity (though this is certainly changing during the Covid 19 pandemic).
Materials and Methods
Lines 74-5 – Do not split the verb. Revised to read “Details of recruitment have been published previously”
Dietary Intake
Lines 83-90 – As mentioned in the abstract of food frequency questionnaire (FFQ) assisting of just 24 questions is extraordinarily short. What is the purpose of the instrument. It isn’t possible to compute meaningful nutrient data from such an instrument. From the description, it appears as though only frequency, and not portion size, was measured.
Lines 95-6 – What is comparator used for assessing reliability in the cited study (21).
Lines 98-103 – How much variation exists within these DAs?
Neighbourhood environment
Nice description of the methods used. Also the PCA approach seems reasonable.
Lines 118-9 – Why were DAs were categorized into tertiles based on material deprivation and social deprivation? With a sample size this large why not quartiles or quintiles?
Covariates
Nice description of the methods used. Also the PCA approach seems reasonable.
Line 125 – When used as a proper noun should white and non-white be capitalized?
Line 128 – I assume that all of the dollar amounts are in Canadian dollars. Please confirm.
Statistical analysis
Line 142 – BMI was not self-reported. BMI was based on self-reported weight and height.
Line 149 – I suggest replacing “Since” with “Because” as the latter does not imply temporality. This could be reworded as “Because there were too few participants within ….”
Line 155 – Specifically, how were the plots analyzed?
Lines 158-62 – I suggest moving the list of variables controlled ahead of the method used to read “Potential Confounders, including age, etc…….”
Lines 163-5 – I am confused. I thought that some of these variables were subject to backward elimination. Are the authors saying that all of these in this list were included based on those criteria? It’s fine if they were. We just need to now.
Line 166 – It is good to know that they had information on individual income and education as estimators of SES.
Results
Line 171 – I believe that they mean Table 1 not “Error! Reference source not found..” However, I think that the mistake may be a bit more complicated.
Line 175 – Replace “had” with “earned”
Line 177 – Replace “Less” with “Fewer” as the subject of the sentence are participants.
Lines 180-1 – The prevalence rate of diabetes seems really low. This is especially concerning given the fact that nearly 70% of the population is either overweight or obese. While the age of the population is such that one might expect to see a bit lower rate than in the general adult population, 5.1% seems unrealistically low.
Table 1. I know what the provinces are. But most readers will not. There’s plenty of room. Spell them out
Line 189 – Maintain level of precision throughout the parenthesis; in this one sentence alone there are 3 different levels of precision in terms of the diet quality score.
Line 209 – Error!
Table 2. There seemed to be much of a difference in diet quality scores by levels of deprivation.
Table 3. Despite what we saw in Table 2, it appears that the unadjusted difference which, presumably, was based on data shown in Table 2. was significant in the unadjusted analysis. This seems to be a large number (i.e. sample size) problem that does not persist after adjusting for important covariates.
Lines 238-40 – What does a difference of 0.7 units mean on a scale of 60.0 units?
Discussion
Lines 251-3 – The authors have, astutely, chosen to hedge in their wording. Clearly, there is no statistically significant difference in the tiny numbers we have seen also raise issues regarding public health significance. However, given that we don’t know the characteristics of the diet quality measure it’s really ultimately hard to tell if there might be clinical relevance for differences on the order of 1 point on the scale.
Lines 260-2 – This is helpful to put the score into context. However, I am a bit skeptical. Are the authors really saying that a shift of .7 is equivalent to almost one serving of fruit and vegetables per day? What would be the effect of, say, a 20 or 30 point shift?
Line 265 – “was based on a nationally representative sample,”
Line 277 – “also may have”
Line 280 – Insert comma before “which”
Lines 301-5 – Among the weaknesses the study should be listed the dietary assessment method. We know very little about this and these kinds of instruments are well known to be associated with bias. The shorter and more obvious, the more likely it is that bias will plague the data. See:
- Hebert JR, Clemow L, Pbert L, Ockene IS, Ockene JK. Social desirability bias in dietary self-report may compromise the validity of dietary intake measures. Int J Epidemiol 1995;24:389-98.
- Hebert JR, Ma Y, Clemow L, Ockene IS, Saperia G, Stanek EJ, Merriam PA, Ockene JK. Gender differences in social desirability and social approval bias in dietary self report. Am J Epidemiol 1997;146:1046-55.
Author Response
Reviewer 2
GENERAL COMMENTS
This is a well-written paper on an interesting subject. My main concern is that it looks as though the differences observed in diet quality score are really small. However we don’t know what the scores actually mean. We do know that the maximum is 60. However virtually all of the averages hover around 38-9 with a standard deviation of around 8.5 (Table 2.). Authors need to do a better job of putting their diet quality measure into meaningful context. It is not easy to draw comparisons with other studies based on this unique measure. So, the onus is upon them to explain this clearly. They also need to put this into context in other ways. Canada is a unique country and the maritime provinces (and Newfoundland Labrador) are really unusual in a lot of ways. For example, there are no really large cities in the region. I believe that Halifax is the largest city with a population of about 400,000. Still, with a total population of around 2 million in a really large area it may be classified as fairly urban! So, generalizing to places with huge cities and lots of urban blight may be difficult. It seems that this may be the underlying cause of what I perceived to be unintuitive results.
-We previously provided context around the scoring scheme in the Discussion and referred to a prior publication which developed the score in the Methods to avoid redundancy. However, the reviewer’s point about needing additional context is well taken and we expanded upon the diet assessment and meaning of the score in the Methods section “The Atlantic PATH diet quality score was developed following methods of the American and Canadian HEIs whose scoring schemes reflect national dietary guidelines and population-based studies for disease prevention (22,23). Intake of food from four food groups (fruits and vegetables, grains, dairy products, meat and alternatives) are assigned a score ranging from 0 to 10. A score of 10 indicates meeting dietary guidelines for a given food group (e.g.≥ 2 servings/day of milk and dairy products for males and females aged ≤ 50 or ≥3 servings/day for males and females aged >50). Intake of foods to consume in moderation (i.e. snacks, desserts and non-diet drinks) and dietary behaviors are assigned a score of 0 or 1 for meeting or not meeting a given recommendation. For example the component ‘make at least half of grain products whole grain each day’ receives a score of 1 if a participants ratio of whole grain to total grain product intake/day is ≥0.5. The components are then summed to generate an overall diet quality score from 0 to 60. Higher scores indicate better diet quality.”.
The reviewer is correct in that the Atlantic provinces and Canada can be considered to be unique in that most areas have relatively low population density. We applied definitions of urbanicity from Canadian Census metrics (census metropolitan areas with populations greater than 10,000 were considered urban). There are thus multiple urban areas in our study. We did not attempt to generalize our findings to huge cities with urban blight. We only suggest that our findings may be generalizable to the Maritime region. We also took great care in describing comparisons to other studies. For example “As outlined above, there are few studies to draw comparisons to and ecological perspectives on dietary intake in Canada which has unique individual, social, environmental, and public policies are even fewer (38)”. We did draw a parallel to the SPOTLIGHT study which included neighbourhoods in 5 urban regions in Europe, 1 of which (Ghent) has a population ~1/2 of Halifax. We have, however, provided additional context on the population and urban/rural nature of the Atlantic provinces to provide additional context to our study context. “It is also important to describe the geographical region of the Atlantic Provinces to provide context to our findings, and details which will help future studies determine if our findings are applicable to them. In 2012, the total population of the Atlantic Provinces was ~2.3 million (43). We applied Canadian Census criteria to define urban areas (populations exceeding 10,000), of which there are 20 in the Atlantic Provinces, with the largest metropolitan area (Halifax) comprising ~400,000 people. ”
SPECIFIC COMMENTS
Abstract
Line 15 – A 24-item FFQ seems very short
-Due to high participant burden and financial constraints associated with administering a more in depth dietary assessment tool (e.g. multiple 24-hr recalls or diet history questionnaires), a 24-item FFQ was used which asked specifically about consumption relative to national dietary guidance from Eating Well with Canada’s Food Guide.
Lines 20-1 – I am not sure what diet quality means in this context. Of course, this applies to text on the lines 23-5 as well.
-We revised the sentence to briefly define diet quality “Diet quality, a metric of how well diet conforms to recommendations was determined….”
Lines 25-6 – Insert hyphen in compound adjective “high-density”
-Done
Introduction
Line 34 – Some might argue that diet is important for many things over and above body weight and body weight itself is determined by physical activity energy expenditure to a large extent.
-We agree that dietary intake is important for more than ‘just’ body weight which is why we also included the importance of diet for prevention of chronic disease and overall health.
Line 39 – Insert hyphen in compound adjectives “nutrient-poor” and “energy-dense”
-Done
Line 41 – I don’t see the distinction between dietary intake and food choices.
-Dietary intake reflects what an individual consumes. Food choices or food selection reflects motivating factors that inform decisions on food for purchase or consumption. We have clarified that these are two distinct aspects of nutrition in the text.
Lines 42-3 – It seems that there are more than one neighborhood environments that must be considered. People may not live where they work or engage in physical activity (though this is certainly changing during the Covid 19 pandemic).
-We agree. The sentence you refer to provides a definition of neighbourhood environment. In our study we did not have information on where an individual works or engages in activity, which we highlight as a limitation in the discussion “This approach may not capture where people engage in daily activities (e.g. work, school, recreation) which may also influence dietary behavior”. That said, it is likely less of a confounder in our study population which as the reviewer (rightly) pointed out does not encompass huge cities where long commutes to work far from home is typical.
Materials and Methods
Lines 74-5 – Do not split the verb. Revised to read “Details of recruitment have been published previously”
-Revised as suggested
Dietary Intake
Lines 83-90 – As mentioned in the abstract of food frequency questionnaire (FFQ) assisting of just 24 questions is extraordinarily short. What is the purpose of the instrument. It isn’t possible to compute meaningful nutrient data from such an instrument. From the description, it appears as though only frequency, and not portion size, was measured.
-We have expanded upon our description of the FFQ which was semi-quantitative. “Participants reported their usual dietary intake using a short, semi-quantitative food frequency questionnaire (FFQ) developed for the Atlantic PATH cohort. The FFQ was designed to capture consumption reflecting the national dietary guidelines at the time of the study (Eating Well with Canada’s Food Guide). The set of 24 questions assessed the daily intake of 60 food items or groups including fruit and vegetables…”
Lines 95-6 – What is comparator used for assessing reliability in the cited study (21).
-Reliability in the study cited in ref 21, refers to the standardized Cronbach’s alpha, a measure of internal consistency and a measure of scale reliability. We have clarified in the text this refers to internal validity
Lines 98-103 – How much variation exists within these DAs?
-We assume the reviewer is referring to variation in neighbourhood environment characteristics, of which there is little variation as a DA has a compact shape approximating a city block. We have provided additional context as to the definition of a DA: “DAs contain between 400 to 700 persons, and are composed of one or more adjacent dissemination block (an area equivalent to a city block).”
Neighbourhood environment
Nice description of the methods used. Also the PCA approach seems reasonable.
-Thank you
Lines 118-9 – Why were DAs were categorized into tertiles based on material deprivation and social deprivation? With a sample size this large why not quartiles or quintiles?
-We did explore use of smaller categories for DA’s. However, the deprivation profiles are created for each of the respective populations in the provinces and then applied to the population in the Atlantic PATH. As a result, smaller categories could result in few participants from the Atlantic PATH (e.g. few in the most deprived area Q5, reflecting the volunteer nature of the study cohort). As a result, tertiles were used.
Covariates
Nice description of the methods used. Also the PCA approach seems reasonable.
Line 125 – When used as a proper noun should white and non-white be capitalized?
-We have edited as suggested
Line 128 – I assume that all of the dollar amounts are in Canadian dollars. Please confirm.
-We have confirmed in the text
Statistical analysis
Line 142 – BMI was not self-reported. BMI was based on self-reported weight and height.
-Corrected to read “body mass index (BMI) calculated from self-reported height and weight”
Line 149 – I suggest replacing “Since” with “Because” as the latter does not imply temporality. This could be reworded as “Because there were too few participants within ….”
-Corrected as suggested
Line 155 – Specifically, how were the plots analyzed?
-Edited to read “Assumptions of linearity in models were confirmed through residual plots that showed random dispersion”
Lines 158-62 – I suggest moving the list of variables controlled ahead of the method used to read “Potential Confounders, including age, etc…….”
-Done
Lines 163-5 – I am confused. I thought that some of these variables were subject to backward elimination. Are the authors saying that all of these in this list were included based on those criteria? It’s fine if they were. We just need to now.
-Yes, these were the variables that met criteria using backward elimination. We have clarified this “Significant confounders identified with backward elimination included age, sex, household income, education, ethnicity, smoking status, alcohol consumption, physical activity, cancer, diabetes, and province of residence, which were included in all models.”
Line 166 – It is good to know that they had information on individual income and education as estimators of SES.
-We agree it’s a strength of the study
Results
Line 171 – I believe that they mean Table 1 not “Error! Reference source not found..” However, I think that the mistake may be a bit more complicated.
-Please see the response to reviewer 1. This reflects an error in the conversion of the word document to pdf. We believe it has now been remedied
Line 175 – Replace “had” with “earned”
-Done
Line 177 – Replace “Less” with “Fewer” as the subject of the sentence are participants.
-Done
Lines 180-1 – The prevalence rate of diabetes seems really low. This is especially concerning given the fact that nearly 70% of the population is either overweight or obese. While the age of the population is such that one might expect to see a bit lower rate than in the general adult population, 5.1% seems unrealistically low.
-We respectfully disagree. Provincial estimates of diagnosed diabetes between 2007-2008 show that the prevalence of diabetes in the Atlantic Provinces among men and women aged 45-64 ranged from 6.9% in Prince Edward Island to a ‘high’ of 11.1% in New Brunswick. Given that the Atlantic PATH is a volunteer cohort which tend to be healthier than the general population, we believe 5.1% is a reasonable prevalence.
Table 1. I know what the provinces are. But most readers will not. There’s plenty of room. Spell them out
-Done
Line 189 – Maintain level of precision throughout the parenthesis; in this one sentence alone there are 3 different levels of precision in terms of the diet quality score.
-We have edited as suggested
Line 209 – Error!
-This was also an error in the manuscript submission system and we believe has been remedied
Table 2. There seemed to be much of a difference in diet quality scores by levels of deprivation.
Table 3. Despite what we saw in Table 2, it appears that the unadjusted difference which, presumably, was based on data shown in Table 2. was significant in the unadjusted analysis. This seems to be a large number (i.e. sample size) problem that does not persist after adjusting for important covariates.
Lines 238-40 – What does a difference of 0.7 units mean on a scale of 60.0 units?
-Please see our earlier reply and context added to the method on scoring
Discussion
Lines 251-3 – The authors have, astutely, chosen to hedge in their wording. Clearly, there is no statistically significant difference in the tiny numbers we have seen also raise issues regarding public health significance. However, given that we don’t know the characteristics of the diet quality measure it’s really ultimately hard to tell if there might be clinical relevance for differences on the order of 1 point on the scale.
Lines 260-2 – This is helpful to put the score into context. However, I am a bit skeptical. Are the authors really saying that a shift of .7 is equivalent to almost one serving of fruit and vegetables per day? What would be the effect of, say, a 20 or 30 point shift?
-Please see our response earlier and additional information provided on scoring. The scoring scheme of the diet quality score was published previously. The effect of a 20 or 30 point shift could manifest in several weighs as some foods/food groups contribute up to 10 units versus 1 unit for choices related to moderation (e.g. having meat alternatives). As an illustrative example, a 20 unit difference in score could reflect a participant who consumed zero servings of fruits and vegetables and zero servings of milk and dairy versus a participant who met the Eating Well with Canada’s Food Guide recommendation for fruits and vegetables ( ≥ 7 servings/day for females and ≥ 8 servings for males Aged ≤ 50 yr and ≥ 7 servings/day for males and females Aged >50) and met the recommendations for milk and dairy products (≥ 2 for males and females Aged ≤ 50 yr and ≥3 for males and females Aged >50).
Line 265 – “was based on a nationally representative sample,”
-Edited as suggested
Line 277 – “also may have”
-Edited as suggested
Line 280 – Insert comma before “which”
-Edited as suggested
Lines 301-5 – Among the weaknesses the study should be listed the dietary assessment method. We know very little about this and these kinds of instruments are well known to be associated with bias. The shorter and more obvious, the more likely it is that bias will plague the data. See:
- Hebert JR, Clemow L, Pbert L, Ockene IS, Ockene JK. Social desirability bias in dietary self-report may compromise the validity of dietary intake measures. Int J Epidemiol 1995;24:389-98.
- Hebert JR, Ma Y, Clemow L, Ockene IS, Saperia G, Stanek EJ, Merriam PA, Ockene JK. Gender differences in social desirability and social approval bias in dietary self report. Am J Epidemiol 1997;146:1046-55.
-We agree that self-reported dietary assessment methods have bias, but the shorter and more obvious does not necessarily mean that it is more bias, depending on the intention of the instrument. For example, 24-hr dietary recalls are one of the more accurate tools for assessing episodically consumed foods. Indeed, each self-reported tool is associated with bias. Our diet tool, while imperfect (like all tools) does abide by recent recommendations that self-report tool should not be used to estimate absolute energy intake due to the extent of misreporting. Rather, self-reported tools are more appropriate for measuring dietary patterns, diet quality scores and components with less error (e.g. fruits and vegetables) (see Kirkpatrick et al. 2017, https://doi.org/10.3945/an.116.014027). We also follow ‘best practices’ by providing details of psychometric testing (validity, reliability). We have added to the weaknesses of the study as suggested “. A further strength was consideration of diet quality; how well an individual follows Canadian dietary recommendations, versus focusing on single foods (e.g. fruits and vegetables) may be more reflective of overall health. However, comparability with other studies is limited by the diet assessment tool which was developed specifically for the Atlantic PATH cohort. Future research which draws on more widely used diet quality measures such as the HEI (41)would facilitate comparisons, although such an undertaking which requires a more in depth dietary assessment in a large population like the Atlantic PATH would be substantial. There are also inherent errors (e.g. underreporting) in self-report intake assessments that may have biased our results toward the null. Although, measurement of diet components have less error than estimates of absolute energy intake (42), which we did not consider in our study.”

Round 2
Reviewer 1 Report
I believe that authors have dealt with all reviewer comments in an appropriate and meaningful fashion,